# Two Innovative Approaches to Optimize Vancomycin Dosing Using Estimated AUC after First Dose: Validation Using Data Generated from Population PK Model Coupled with Monte-Carlo Simulation and Comparison with the First-Order PK Equation Approach

**DOI:** 10.3390/pharmaceutics14051004

**Published:** 2022-05-07

**Authors:** Qingxia Liu, Huiping Huang, Baohua Xu, Dandan Li, Maobai Liu, Imam H. Shaik, Xuemei Wu

**Affiliations:** 1Department of Pharmacy, Fujian Medical University Union Hospital, Fuzhou 350001, China; qingxialiu@fjmu.edu.cn (Q.L.); huanghuiping@fjmu.edu.cn (H.H.); xunabell@fjmu.edu.cn (B.X.); dandanli@fjmu.edu.cn (D.L.); liumb0591@fjmu.edu.cn (M.L.); 2School of Pharmacy, Fujian Medical University, Fuzhou 350001, China; 3Department of Pharmaceutical Sciences, School of Pharmacy, University of Pittsburgh, Pittsburgh, PA 15260, USA; ihs4@pitt.edu

**Keywords:** vancomycin, dose optimization, pharmacokinetics/pharmacodynamics, population pharmacokinetics, Monte Carlo simulation

## Abstract

The revised consensus guidelines for optimizing vancomycin doses suggest that maintaining the area under the concentration-time curve to minimal inhibitory concentration ratio (AUC/MIC) of 400–600 mg·h/L is the target pharmacokinetic/pharmacodynamic (PK/PD) index for efficacy. AUC-guided dosing approach uses a first-order pharmacokinetics (PK) equation to estimate AUC using two samples obtained at steady state and one-compartment model, which can cause inaccurate AUC estimation and fail to achieve the effective PK/PD target early in therapy (days 1 and 2). To achieve an efficacy target from the third or fourth dose, two innovative approaches (Method 1 and Method 2) to estimate vancomycin AUC at steady state (AUC_SS_) using two-compartment model and three or four levels after the first dose are proposed. The feasibility of the proposed methods was evaluated and compared with another published dosing algorithm (Method 3), which uses two samples and a one-compartment approach. Monte Carlo simulation was performed using a well-established population PK model, and concentration-time profiles for virtual patients with various degrees of renal function were generated, with 1000 subjects per group. AUC extrapolated to infinity (*AUC*_0–∞_) after the first dose was estimated using the three methods, whereas reference AUC (*AUC_ref_*) was calculated using the linear-trapezoidal method at steady state after repeated doses. The ratio of *AUC*_0–∞_: *AUC_ref_* and % bias were selected as the indicators to evaluate the accuracy of three methods. Sensitivity analysis was performed to examine the influence of change in each sampling time on the estimated *AUC*_0–∞_ using the two proposed approaches. For simulated patients with various creatinine clearance, the mean of *AUC*_0–∞_: *AUC_ref_* obtained from Method 1, Method 2 and Method 3 ranged between 0.98 to 1, 0.96 to 0.99, and 0.44 to 0.69, respectively. The mean bias observed with the three methods was −0.10% to −2.09%, −1.30% to −3.59% and −30.75% to −55.53%, respectively. The largest mean bias observed by changing sampling time while using Method 1 and Method 2 were −4.30% and −10.50%, respectively. Three user-friendly and easy-to-use excel calculators were built based on the two proposed methods. The results showed that our approaches ensured sufficient accuracy and achieved target PK/PD index early and were superior to the published methodologies. Our methodology has the potential to be used for vancomycin dose optimization and can be easily implemented in clinical practice.

## 1. Introduction

Recently approved consensus guidelines and inhouse protocols in clinics guide vancomycin use in clinical management of *Methicillin-Resistant Staphylococcus Aureus* (MRSA) infections [1,2,3]. Vancomycin is a glycol-peptide antibiotic and shows significant inter-patient pharmacokinetic (PK) variability and dose-dependent nephrotoxicity and ototoxicity. Therapeutic Drug Monitoring (TDM) is recommended for vancomycin to tailor the most appropriate dose for specific patients [4,5] to achieve clinical benefit and avoid toxicity. Vancomycin exhibits a long post antibiotic effect (PAE) and the area under the concentration-time curve to minimal inhibitory concentration ratio (AUC/MIC) has been identified as the suitable pharmacokinetic/pharmacodynamic (PK/PD) target parameter for vancomycin TDM [6]. International consensus guideline developed in 2009 suggested a trough level of 15–20 mg/L as the surrogate to optimize the dose [3]. Single-center, observational studies and retrospective analysis of data showed poor relationship between AUC and trough level of vancomycin [7]. The newly revised guidelines emphasized maintaining AUC of 400–600 mg·h/L daily as the new PK/PD index assuming vancomycin MIC at 1 mg/L [8].

The revised guideline recommends two approaches to achieve the target for AUC-guided dosing using (1) first-order PK equation to estimate the AUC based on two concentrations sampled at 1–2 h after infusion and trough concentrations before the next dose and (2) a population pharmacokinetic (pop PK) modelling coupled with Bayesian derived AUC monitoring based on the 1 or 2 concentrations with at least 1 sample at trough level [8,9,10]. The pop PK and Bayesian statistical methods approach has the advantage of accurately estimating AUC in a specific patient using limited sampling strategies [11,12]. Therefore, it is preferable to follow AUC-guided dosing for vancomycin. However, not all the clinical programs practicing TDM for vancomycin have access to Bayesian software or afford the cost of license for commercial Bayesian forecasting programs which use either one- or two-compartment PK models with different approaches to incorporate covariates. Additionally, there is confusion among clinicians and pharmacists while choosing suitable software for designing a personalized dosing regimen for the patients [13]. It is also impractical to implement these programs without proper training about application of models and interpretation of outcomes of the software programs [2,14,15]. Moreover, using pop PK models which are incapable of representing the actual patients with different clinical conditions (e.g., renal failure, sepsis, cystic fibrosis) and obtaining AUC estimations using the inappropriate Bayesian software maybe less accurate and may not achieve projected therapeutic outcomes [16]. 

In the absence of licensed software programs, first-order PK estimation guided dosing for vancomycin can be implemented for the ease of calculations. However, the reported PK studies indicate that the disposition of intravenously administered vancomycin is characterized by two phases. The rapidly declining phase after infusion, which shows a (distribution) half-life (t_1/2α_) of 0.5 to 1 h and the terminal elimination half-life (t_1/2β_) ranges from 3 to 9 h in subjects with normal renal function [17]. It is widely accepted that PK of vancomycin is best described by two-compartment model [13,18,19]. Apparently, the ability to accurately predict the concentrations is superior with a two-compartment model as compared to a one-compartment model. The one compartment model can introduce a large bias in AUC calculation from PK profiles by ignoring a slope of steep distribution phase [20]. Therefore, two sample concentrations recommended in the previous guideline may not provide accurate estimation of AUC for vancomycin and doses selected from the inaccurate estimations may fail to produce the expected therapeutic outcomes. Furthermore, in the first-order PK-guided AUC estimation, both the samples are preferred to be around the steady state and the earliest samples could be collected after 2 days of treatment to reach steady state in patients with normal renal function. It will take much longer to reach steady state in patients with renal impairment due to the prolonged the elimination half-life. The delayed TDM and dosing adjustment may lead to the treatment failure or toxicity.

To adjust vancomycin dosing at an early time, researchers ventured to calculate the personalized PK parameters using two vancomycin concentrations sampled after the first dose and guide the subsequent dosing regimen [21]. However, AUC estimated using parameters based on the one-compartment model is not plausible. 

The aim of the current study is to evaluate two innovative approaches, using three or four concentrations sampled after the first dose, to calculate PK parameters using the two-compartment model and guide the subsequent dosing adjustments from the third dose. The PK profiles of 1000 virtual patients with normal renal function generated from a well-established pop PK model using Monte Carlo simulation were used to validate the feasibility of the proposed approaches. The predictive accuracy is compared with that of the previous reported methodology [21]. Furthermore, the proposed methodologies were also tested in virtual patients with different levels of renal impairment.

## 2. Methods

### 2.1. Literature Review and Model Selection

A comprehensive literature search for pop PK analyses of vancomycin was carried out in the PubMed database up to June 2021 from its inception. Key words “vancomycin”, “population pharmacokinetic” or “population pharmacokinetics” or “non-linear mixed effect” or “Bayes” or “pharmacokinetic model” or “NONMEM” or “pop PK” or “PPK” were used as the search terms. Moreover, a thorough inspection of pertinent reference lists of selected papers was conducted to identify any additional relevant published reports. The search was limited to English language.

Titles and abstracts were reviewed, the publications with irrelevant topics or pop PK models for vancomycin described by one-compartment model or three-compartment model were excluded. In the published pop PK studies with two-compartment structural model, the model developed with data from more patients as well as more maximum samples per patient was selected to generate the virtual PK profiles. 

### 2.2. Monte Carlo Simulation of PK Profile in Patients with Normal Renal Function

The selected pop PK model of vancomycin was re-coded, in which the creatinine clearance (CLcr) was incorporated as the covariate of clearance (CL). A Monte Carlo simulation was performed using NONMEM software (version 7.5) to generate simulated concentration time profiles for 1000 virtual patients with normal renal function (CLcr value: 100 mL/min). According to the package insert for vancomycin, simulations were performed to generate plasma concentration profiles for intravenous infusion of 1000 mg vancomycin every 12 h in virtual patients. The infusion time was set for 60 min. All the patients were assumed to have reached steady state on day 15 (the twenty-ninth dose). The concentration of vancomycin was simulated every half an hour in the first and the twenty-ninth dosing interval. 

### 2.3. Estimation of AUC Using Four Serum Concentrations (Method 1)

After the first dose, the concentrations at t_1_, t_2_, t_3_, and t_4_ (t_1_ = 1.5 h, t_2_ = 2.5 h, t_3_ = 9.5 h, t_4_ = 11.5 h) were selected to calculate the AUC extrapolated to infinity (*AUC*_0–∞_) using the two-compartment PK model equation.

For two-compartment PK model, the change in concentration over time after completion of intravenous infusion can be calculated using the following equation:(1)$$C=\frac{k_{0}\left(\alpha -k_{21}\right)\left(1-e^{-\alpha T}\right)}{V_{C}\alpha \left(\alpha -\beta \right)}e^{-\alpha t^{\prime }}+\frac{k_{0}\left(k_{21}-\beta \right)\left(1-e^{-\beta T}\right)}{V_{C}\beta \left(\alpha -\beta \right)}e^{-\beta t^{\prime }}$$
where *V_C_* is the volume of distribution in the central compartment, *k*_21_ is intra-compartmental rate constant, *α* and *β* are rate constant for distribution and elimination phases, respectively, *t*′ is the time between *C* and end of the infusion, *Τ* is the infusion time.

For the convenience of calculation, the above equation can be simplified by replacing the complex portions of the formula with two parameters of *R* and *S*.
(2)$$R=\frac{k_{0}\left(\alpha -k_{21}\right)\left(1-e^{-\alpha T}\right)}{V_{C}\alpha \left(\alpha -\beta \right)}$$
(3)$$S=\frac{k_{0}\left(k_{21}-\beta \right)\left(1-e^{-\beta T}\right)}{V_{C}\beta \left(\alpha -\beta \right)}$$

Hence, the equation is simplified to:(4)$$C=Re^{-\alpha t^{\prime }}+Se^{-\beta t^{\prime }}$$

Using the method of residuals, the bi-exponential function can be segmented into two mono-exponential functions. The rate of decrease in the plasma concentration during distributed phase is much higher than the rate of decrease in the plasma concentration during elimination phase, where *α* ≫ *β*. 

When *t* is adequately large (terminal portion of curve), *Re*^−*αt*^^′^ is more quickly approaching zero, so the equation at the terminal portion of the curve can be simplified to:(5)$$C^{\prime }=Se^{-\beta t^{\prime }}$$

The terminal linear portion is extrapolated towards distribution phase and the extrapolated concentration value (*C*′) at the distribution phase of the curve at each corresponding time can be calculated according to Equation (5), then the residual concentration (*Cr*) is calculated with the following equation:(6)$$Cr=C-C^{\prime }=Re^{-\alpha t^{\prime }}$$

The parameters for *β*, *S* and *α*, *R* are obtained using Equations (5) and (6), respectively (Figure 1A). *AUC*_0–∞_ is calculated using 2 approaches (Figure 2).

The first approach uses the relevant parameters for two-compartment PK model following intravenous injection and calculate *AUC*_0–∞_. The relationship between the parameters for *S* and *R* after intravenous infusion and the zero-time intercepts *A* and *B* after intravenous injection are as follows [22]:(7)$$A=\frac{\alpha T}{\left(1-e^{-\alpha T}\right)}R$$
(8)$$B=\frac{\beta T}{\left(1-e^{-\beta T}\right)}S$$

*AUC*_0–∞_ was calculated using the following equation:(9)$$AUC_{0-\infty \;}=\frac{A}{\alpha }+\frac{\;B}{\beta }$$

The second approach is to calculate the *AUC*_0–∞_ via the relevant formula for intravenous infusion using two-compartment PK model.

The parameters for *k*_21_ and *V_C_* can be calculated by Equations (2) and (3). The following equations can be used to calculate the *AUC*_0–∞_.
(10)$$\alpha \beta =k_{21}k_{10}$$
(11)$$CL\;=k_{10}V_{C}$$
(12)$$AUC_{0-\infty }=\frac{X_{0}}{CL}$$
where *k*_10_ is the elimination rate constant from the central compartment, *X*_0_ is the initial dose.

### 2.4. Estimation of AUC Using Three Serum Concentrations (Method 2)

Another method to estimate *AUC*_0–∞_ was using the linear-trapezoidal method based on three serum concentrations at *t*_1_, *t*_2_, and *t*_3_ (*t*_1_ = 1 h, *t*_2_ = 9.5 h, *t*_3_ = 11.5 h) after the first dose of vancomycin (Figure 1B). With this method, an extrapolated concentration at 3 h after beginning vancomycin infusion is needed to ensure the accuracy of *AUC*_0–∞_.

First, the elimination rate constant (*β*) was calculated from the terminal portion of the curve using the following equation: (13)$$\beta =\;\ln\left(C_{2}/C_{3}\right)\;/\;(t_{3}-t_{2})$$

S was calculated by using the formula in Figure 2 step 1. 

The extrapolated concentration *C_e_* at *t_e_* (*t_e_* = 3 h) was calculated using the Equation (5).

*AUC*_0–∞_ was calculated with a linear-trapezoidal method using the following equation:(14)$$AUC_{0-\infty }=0.5C_{1}t_{1}+0.5\left(C_{1}+C_{e}\right)\left(t_{e}-t_{1}\right)+0.5\left(C_{e}+C_{2}\right)\left(t_{2}-t_{e}\right)+0.5\left(C_{2}+C_{3}\right)\left(t_{3\;}-t_{2}\right)+\frac{C_{3}}{\beta }$$

### 2.5. Estimation of AUC Using Two Serum Concentrations (Method 3)

According to the method described by Flannery et al. [21], the concentration-time profiles of simulated patients were assumed to fit a one-compartment model. Two concentrations of each subject, i.e., levels at t_1_ (2.5 h) and t_2_ (11.5 h) after the first dose, were used to calculate the *AUC*_0–∞_ (Figure 1).

For the one-compartment model, the change in plasma concentration during the intravenous infusion can be described with the equation: (15)$$C=\frac{k_{0}}{Vk}(1-e^{-kt})$$
where *k*_0_ is the infusion rate constant, *V* is the apparent volume of distribution, *k* is the elimination rate constant, and *t* is the time after starting infusion.

Changes in the concentration over time after completion of the intravenous infusion were calculated by the equation:(16)$$C=C_{0}e^{-kt^{\prime }}$$
where *C*_0_ is the concentrations at the end of infusion, and $t^{\prime }$ is the time between *C* and end of the infusion.

The method of *AUC*_0–∞_ calculation was based on the Equations (15) and (16).

The specific calculation steps are as follows:Step 1: Estimation of elimination rate constant (*k*)
(17)$$k=\frac{Ln\left(\frac{C_{1}}{C_{2}}\right)}{\left(t_{2}-t_{1}\right)}$$

Step 2: Estimation of concentration at the end of infusion (*C_max_*)


(18)
$$C_{max}=\frac{C_{1}}{e^{-\left(t_{2}-t_{1}\right)}}$$


Step 3: Calculation of volume of distribution (*V_d_*)


(19)
$$V_{d}=\frac{X_{0}}{T}\times \frac{\left(1-e^{-kT}\right)}{kC_{max}}$$


Step 4: Calculation of clearance (*CL*)


(20)
$$CL=kV_{d}$$


Step 5: Calculation of *AUC*_0–∞_


(21)
$$AUC_{0-\infty }=\frac{X_{0}}{CL}$$


### 2.6. Method Evaluation

Given that *AUC*_0–12_ at steady state is reflective of the exposure during a dose interval, it is theoretically equal to *AUC*_0–∞_ after the first dose. *AUC*_0–12_ of the twenty-ninth dose calculated using the linear-trapezoidal method was defined as the reference AUC (*AUC_ref_*). The feasibility of the proposed approaches was evaluated by comparing *AUC*_0–∞_ values after the first dose with *AUC_ref_* in the simulated population.

The mean ratio of calculated *AUC*_0–∞_ after the first dose to *AUC_ref_* (*AUC*_0–∞_: *AUC_ref_*) was defined as one of the indicators to reflect the accuracy of the three methods. Bias and relative root mean square error *RMSE* (%) are calculated as criteria for judging accuracy.
(22)$$\mathrm{Bias}\;(\%)=\frac{AUC_{0-\infty }-AUC_{ref}}{AUC_{ref}}\times 100$$
(23)$$RMSE\;\left(\%\right)=\sqrt{\frac{1}{N}\sum_{i=1}^{N}{\left(\frac{AUC_{0-\infty ,i}-AUC_{ref,i}}{AUC_{ref,i}}\right)}^{2}}\times 100$$

### 2.7. Extrapolation to Patients with Various Degrees of Renal Impairment and Method Evaluation

To evaluate the accuracy of different approaches in patients with various degrees of renal function, virtual patients with CLcr of 60, 50, 35, and 25 mL/min were simulated with 1000 subjects in each group. Elimination half-life was relatively longer in patients with renal impairment, the dosing regimen and the time to reach steady state were varied across subjects with different CLcr. The more severe the renal impairment is, the longer it will be to reach steady state, and a lower daily dose would be needed to maintain therapeutically relevant target AUC. In patients with CLcr values of 35 and 25 mL/min, the dosing interval was adjusted to 24 h and the sampling time was delayed accordingly, while using Method 1 and Method 2. The detailed dosing regimen and related calculation conditions are presented in Table 1. AUC within the last dose interval calculated using the linear-trapezoidal method was defined as *AUC_ref_*.

### 2.8. Sensitivity Analysis

Method 1 and Method 2 are highly recommended due to low bias. A sensitivity analysis was performed to examine the influence of shifting each sampling time on *AUC*_0–∞_. While utilizing Monte Carlo simulation, the concentration of vancomycin of each patient can be obtained at any time point, so the minimum change of sampling time was set to half an hour or one hour, depending on the dosing interval. *AUC*_0–∞_ of simulated patients was calculated by changing one scheduled sampling time while others remained unchanged in all the scenarios. Bias was calculated across all the scenarios. 

### 2.9. Development of the Dose Calculator Tool

We built three dose calculators using Microsoft Excel based on the PK equations for Method 1 and Method 2 approaches. The equations were embedded in the Microsoft Excel. The subsequent dosing regimen for the 3rd or 4th dose can be adjusted automatically after inputting the patient information, sampling times and their corresponding plasma vancomycin levels.

## 3. Results

### 3.1. Literature Review and Model Selection 

After a systematic literature search, 125 pop PK models were identified in publications, which described vancomycin PK with either one-(58.4%, 73/125), two-(34.4%, 43/125) or three-(7.2%, 9/125) compartment structural models. Most of the studies primarily described vancomycin PK in special sub-populations. A model based on richly sampled vancomycin data from 1253 plasma samples collected from 190 participants was selected [23]. Study was conducted in a Japanese adult population, and more than half were older than 65 years. The average number of samples per patient was 6.6, with samples collected at peak, 1 and 2 h after the end of infusion, and just before the next dose. The model incorporated CLcr as a covariate of CL. 

### 3.2. Monte Carlo Simulation and Method Evaluation

Vancomycin daily exposure at steady state (*AUC*_0–24, ss_) was calculated as 2 times of *AUC_ref_* after the last dose. For patients with normal renal function (CLcr value: 100 mL/min), the probability of target attainments (PTA), proportion of *AUC*_0–24, ss_ between 400 and 600 mg·h/L, was 38.2%, and the percentage of *AUC*_0–24, ss_ above 600 mg·h/L and below 400 mg·h/L was 44.3% and 17.5%, respectively. The mean *AUC_ref_* was 304.87 mg·h/L. *AUC*_0–∞_ after the first dose calculated using the three methods was 300.37, 295.92 and 202.67 mg·h/L, with the mean (±SD) ratio of *AUC*_0–∞_ to *AUC_ref_* for Method 1, Method 2 and Method 3 being 0.99 (±0.02), 0.98 (±0.02) and 0.69 (±0.10), respectively (Figure 3). There are two approaches to calculate *AUC*_0–∞_ after the first dose in Method 1, the calculated value was same using either way. The mean % bias (±SD) using Method 1, Method 2 and Method 3 was −1.12% (±1.46%), −2.40% (±2.46%) and −30.75% (±9.81%), respectively (Figure 3). The RSME for Method 1, Method 2 and Method 3 was 1.84, 3.37 and 32.28%, respectively.

### 3.3. Method Evaluation in Patients with Various Degrees of Renal Impairment 

*AUC*_0–24, ss_ was calculated as two times *AUC_ref_* using 12 h dosing intervals, and changing the dosing interval to 24 h showed that *AUC*_0–24, ss_ was equal to *AUC_ref_* (Figure 3). The dosing regimen and the sampling time schedule in patients with various degrees of renal function is shown in Table 1. In the patients with CLcr of 60, 50, 35, and 25 mL/min, the proportion showing an *AUC*_0–24, ss_ between 400 and 600 mg·h/L were 39.3%, 35.5%, 36.3% and 35.4%, respectively.

The mean *AUC_ref_* in patients with CLcr of 60, 50, 35, and 25 mL/min was 284.59, 227.67, 478.49 and 446.27 mg·h/L, respectively. The *AUC*_0–∞_ estimated after the first dose using Method 1, Method 2 and Method 3 in patients with various CLcr were 278.80, 274.58 and 176.67 mg·h/L (CLcr: 60 mL/min), 221.57, 218.22, 131.74 mg·h/L (CLcr: 50 mL/min), 477.91, 471.92, 234.90 mg·h/L (CLcr: 35 mL/min) and 445.57, 440.84, 181.68 mg·h/L (CLcr: 25 mL/min), respectively.

The mean (±SD) ratio of *AUC*_0–∞_ to *AUC_ref_* calculated using Method 1 were 0.98 (±0.02), 0.98 (±0.02), 1 (±0.00), 1 (±0.00) for patients with CLcr of 60, 50, 35, and 25 mL/min. The RMSE was 2.38, 3.16, 0.19, and 0.39%, respectively. The corresponding values obtained using Method 2 were 0.97 (±0.02), 0.96 (±0.03), 0.99 (±0.11) and 0.99 (±0.01), respectively. The RMSE was 3.80, 4.51, 1.63, and 1.55%, respectively. For Method 3, the mean (±SD) ratio of *AUC*_0–∞_ to *AUC_ref_* was 0.66 (±0.11), 0.62 (±0.12), 0.53 (±0.13) and 0.44 (±0.12) in the respective patient populations. The RMSE was 36.07, 40.03, 48.71 and 56.91%, respectively. Comparing the values obtained for patients with normal renal function, the mean bias of Method 1 was slightly increased in subjects with CLcr of 60 and 50 mL/min (−1.56%, −2.09% vs.−1.12%). While in patients with CLcr of 35 and 25 mL/min, the mean bias was slightly decreased (−0.10% and −0.15% vs.−1.12%). The mean (±SD) bias of Method 2 for patients with CLcr of 60, 50, 35 and 25 mL/min was −2.98% (±2.45%), 3.59% (±2.74%), −1.38% (±0.88%) and −1.30% (±0.86%), respectively, whereas the corresponding bias with Method 3 was −34.35% (±11.01%), −38.24% (±11.85%), −47.07% (±12.52%) and −55.53% (±12.47%). 

### 3.4. Sensitivity Analysis

We also evaluated the effect of sampling time change on the bias by adjusting one scheduled sample time backward or forward while other samples remained unchanged. The bias observed in different scenarios is presented in Figure 4 for Method 1 and Figure 5 for Method 2. The mean bias for Method 1 ranged from −0.56% to −2.43%, −0.79% to −3.28%, −1.08% to −4.3%, −0.05% to −0.20%, and −0.06% to −0.33% in patients with CLcr of 100, 60, 50, 35 and 25 mL/min, respectively. Whereas the corresponding bias values observed with Method 2 ranged from −1.70% to −9.44%, −2.12% to −10.50%, −2.51% to −10.00%, −1.33% to −7.84%, and −1.22% to −6.05% for patients with CLcr of 100, 60, 50, 35 and 25 mL/min, respectively.

### 3.5. Excel Calculator

After obtaining the information after first dose, the sampling time and measured vancomycin levels are entered into Excel calculator and the PK parameters were calculated automatically. The totally daily dose (*TDD*) can be calculated with the following equation:(24)$$TDD=\frac{X_{0}}{AUC_{0-\infty }}\times AUC_{goal}$$

To make it easier for the clinician to adjust the subsequent dose, we embedded the therapeutic window of vancomycin in the calculator, and the margin of *TDD* could be obtained automatically (Figure 6).

## 4. Discussion

To the best of our knowledge, this is the first study to estimate vancomycin *AUC*_0–∞_ using the levels from the first dose based on the mathematical approach using a two-compartment model. There are several noteworthy observations from our studies. First, we were able to estimate AUC of vancomycin at steady state using 3 or 4 samples obtained after the first dose and clinically relevant vancomycin dose adjustment could be achieved latest by the third dose. Compared to the sampling at steady state, which takes up to 3 days, our approach can achieve the PK/PD target quickly. This approach is especially useful in patients with renal impairment considering their long elimination half-life for vancomycin and relatively long time necessary to reach steady state. Second, in comparison with Method 3 that estimating AUC after first dose based on one-compartment model, our proposed methods provide a more accurate estimation. The virtual concentration-time profiles generated using a pop PK model for vancomycin were instrumental in validating our proposed approaches. Both of our proposed methods accurately estimate vancomycin AUC at steady state (AUC_SS_) in patients with various degrees of renal function. Furthermore, sensitivity analysis demonstrated that shifting sampling time had little effect on the final estimates. Based on our results, we proposed the sampling times for patients with various degrees of renal function and dosing interval, and three user-friendly excel calculators for convenient implementation.

The disposition of vancomycin in vivo is a complex process. Though vancomycin concentration-time profiles have been reported as mono-, bi- or triphasic, the most of the literature reports incline towards biphasic process after intravenous administration [24]. In the PK studies involving intensive sampling, 11 studies show that the PK of vancomycin is best fitted to a two-compartment model [18,25,26,27,28,29,30,31,32,33,34]. While Lisby-Sutch et al. [35] reported one-compartment model and Healy et al. and Krogstad et al. [36,37] used a three-compartment model to describe the PK of vancomycin. Being a highly hydrophilic antibiotic, vancomycin extensively distributes into the body fluids and tissues [38] and shows a volume of distribution of 0.2–1.6 L/kg at steady state, which is comparable to volume of total body water [39,40,41].

Majority of the published pop PK studies for vancomycin (73/125) reported using a one-compartment structural model. Most vancomycin pop PK models were derived from data obtained near trough concentrations or concentrations collected after the initial distribution phase, it is reasonable that more studies employed one-compartment analysis for model development [42,43,44]. However, these pop PK models are not sufficient to simulate the PK profiles for entire concentration-time course during a dosing interval.

Several sampling-related factors, such as number of subjects, number of samples per subject, nominal sampling times, and blood sampling after single or multiple doses [20,45], were considered while selecting pop PK model to generate simulated data. We chose a two-compartment PK model based on richly sampled vancomycin data with an average of 6.6 samples per subject. The sampling times were evenly distributed in distribution and elimination phases, and the samples were obtained after single and multiple doses. Therefore, the selected model was developed with input accommodating two-compartment model characteristics of vancomycin. In addition, the model was developed with data obtained from patients with different degrees of renal function, and the CLcr was identified as a significant covariate for CL. Given that vancomycin is not metabolized and approximately 90% of the dose is eliminated by kidney [46], kidney function assessed as CLcr shows significant correlation with vancomycin CL. The effect of body weight on PK parameters is debatable based on results from different studies [47,48,49,50]. Theoretically, an increase in weight may lead to higher clearance and volume of distribution. Data from published studies indicates that the coefficient of variation of body weight is 17.7% (54.3 ± 9.6 kg) is relatively low and shows an even distribution of body weight which may explain its exclusion from covariate while modeling for PK parameters.

Pop PK model estimates both fixed and random effects (the inter- and intra-individual variability) to explain the variability in PK parameters, such as CL and the volume of distribution [51]. Monte Carlo simulation allows the characterization of drug concentration-time profiles for a large number of simulated subjects through repeated and random sampling of data from a multivariate distribution [52]. The simulated data can provide information about the range of drug disposition characteristics in a specific population, which can be further used to determine the probability of obtaining a critical target exposure [53]. In this study, we simulated virtual PK profiles in patients with various degrees of renal function (1000 subjects in per group) to evaluate the feasibility, validity, and applicability of the proposed methods to other patient populations.

Considering relative short elimination half-life (3–9 h) of vancomycin in patients with normal renal function, the current guideline recommends vancomycin TDM at the fourth and fifth doses [54]. Using Monte Carlo simulation, we were able to generate several simulated patients with extremely long elimination half-life due to the large inter-patient variability and simulated up to 29 repeated doses to make sure that all the patients reach steady state. Studies indicated that most of the administered vancomycin was eliminated via kidney and the elimination half-life of vancomycin is longer in patients with renal impairment [19,38]. The simulations were performed with adjusting the number of doses and total simulation time to accommodate for the changes in CLcr in different patient populations.

Our simulations in different populations reiterate the need for TDM for vancomycin therapy. The guideline recommended initial doses of vancomycin for patients with CLcr of 100, 60, 50, 35 and 25 mL/min are presented in Table 1 [55]. The observations show that *AUC_ref_* outside the expected therapeutic window was more than 60%, not only in patients with normal renal function but also in patients with impaired renal function. This clearly indicates that the initial dosing recommendations were not appropriate for most of the patients. Our simulations show that for 82.5% of the patients with normal renal function, *AUC*_0–24, ss_ was above 400 mg·h/L and these observations were consistent with the results reported by Alqahtani et al. [56], but were higher than the values reported by Hamada et al. and Mei et al. [57,58]. Results reported by other researchers clearly indicate that TDM is needed for vancomycin and the unmet need for dosing adjustment methodologies is fulfilled by our approach [43,59].

Method 1 needed four samples with two samples in the distribution phase and two samples in the post distribution phase. For two-compartment model and the method of residuals, the last two samples should be collected in the elimination phase to accurately estimate the elimination rate constant β. Since the distribution half-life of vancomycin was reported to be 0.5–1 h [17], the first two sampling time points at 1.5 h and 2.5 h were plausibly located in the distribution phase. Assuming 12 h dosing intervals, the last two sampling time points were set at 9.5 h and 11.5 h. When dosing interval was increased to 24 h in patients with renal impairment, the corresponding time points were changed to 12 h and 23 h to avoid the influence of distribution on the estimation of β. The first two sampling time points were set at 1.5 h and 2.5 h to consistently place them in the distribution phase.

Considering the approach discussed above, the last two sampling times in Method 2 were same as Method 1. Since *AUC*_0–∞_ was calculated using linear-trapezoidal rule in Method 2, the first sample set at 1 h, namely the time at the end of infusion, was selected to calculate AUC to minimize the bias in AUC estimation. As illustrated in Figure 1B, the plasma concentrations declined rapidly after the infusion is stopped. The vancomycin levels seem to transition from distribution to the elimination phase approximately 2 h after the end of infusion. Using the elimination rate constant obtained from last two points, we estimated the concentration at 2 h after the end of infusion in Method 2. *AUC*_0–∞_ was calculated using linear-trapezoidal rule based on three measured levels at the scheduled times and one extrapolated level.

Method 3 is an adopted approach from Flannery et al. [21], which recommends collecting two samples. One sample at 2.5 h (1–2 h after the end of infusion), and the other level at 11.5 h (trough level) are collected. Patient-specific AUC are calculated based on one-compartment model equations. Several published reports suggested that vancomycin exhibits longer distribution phase [17,60,61] and using one compartment approach may result in inaccurate PK estimations. Flannery et al. [21] reported that PTA at steady state was 58.6% with one compartment approach, which is also in accordance with observations from Ackerman et al. [62], suggesting that the one-compartment model may not be sufficient to predict vancomycin exposure.

The ratio of *AUC*_0–∞_: *AUC_ref_* and corresponding bias from *AUC_ref_* obtained using three methods are presented in Figure 3. The vancomycin *AUC*_0–∞_ calculated using four (Method 1) or three samples (Method 2) was superior to approach utilizing two samples (Method 3). In patients with normal renal function, the estimated *AUC*_0–∞_ using Method 1, Method 2 after the first dose was similar to *AUC_ref_*, with mean ratio of 0.99 and 0.98, respectively, whereas the mean ratio was 0.69 for Method 3. The observation also showed similar trends in patients with various degrees of renal impairment (CLcr values: 60, 50, 35, and 25 mL/min). Among the approaches tested, Method 1 generated best estimates, while Method 3 showed the worst performance (Figure 3). The highest mean bias of Method 1, Method 2 and Method 3 was −2.09%, −3.59% and −55.53%, respectively. The highest RMSE observed for Method 1, Method 2 and Method 3 was 3.16, 4.51, and 56.91%, respectively. There was no significant difference in accuracy and bias between Method 1 and Method 2. Additionally, Method 2 requires fewer samples, so it would be convenient to implement in the clinical practice.

The optimal sampling window for collecting three or four samples is critical while applying our approaches. A sensitivity analysis conducted to identify the most appropriate times for sample collection showed that changing the sampling times has a little influence on the estimated *AUC*_0–∞_ when using Method 1 or Method 2 in patients with various degrees of renal function. As depicted in Figure 4, for Method 1, the bias of *AUC*_0–∞_ against *AUC_ref_* was primarily dependent on the last two sampling points. Comparing the *AUC*_0–∞_ value obtained using the scheduled sampling times (Table 1), the bias showed a slight increase when the last two time points were moved forward. The highest mean bias was −4.30% when the sampling times were set at 1.5, 2.5, 8 and 11.5 h in patients with CLcr of 50 mL/min. The bias was not altered when the first two sampling time points changed. Similarly, while using Method 2, the total bias was slightly altered when the last two sampling times were moved while others remain unchanged (Figure 5), which indicated similar tendency with methods 1 and 2. Therefore, setting the last two sampling time points between 8–12 h is feasible, which could offer more flexibility in clinic. Changing the first sampling time to 0.5 h after starting the infusion substantially increased the mean bias to in patients with CLcr of 60 mL/min. These results indicate that the first sample should be collected after completion of infusion to obtain a more accurate estimate of *AUC*_0–∞_ when using Method 2.

The operation interface of the Excel calculator is straightforward, clear and user friendly. Doctor, pharmacist or any clinician involved in patient care can select any of the three proposed calculators depending on the sample times for appropriate dose selection for vancomycin TDM. Furthermore, the design principle of the calculators can be easily applied for personalized dosing of other drugs following two-compartment model.

Though the proposed approaches provide better estimates for dose selection, a few limitations should be noted. Compared to other methods, the proposed approaches in this study require more samples, which maybe not be convenient for clinical implementation [13]. However, our approaches are worth considering, as the potential benefits outweigh the costs and efforts. Medications such as Busulfan and Mycophenolic acid have similar issues, but are monitored well in clinic with 2–6 samples (Busulfan) or limited sampling strategy with 3 samples (Mycophenolic acid) [63,64]. Close cooperation between clinicians and nurses is critical for efficient achievement of TDM for highly potent and sensitive drugs. The proposed methods were not explored in patients with dosing intervals lower than 12 h and should not be applied in protocols with <12 h dosing intervals. It is worth noting that the high accuracy of our approaches is based on the assumption that vancomycin follows a two-compartment model. In special populations such as critically ill patients, a two-compartment model assumption may not be applicable due to significant physiological changes observed in the patients. The PK parameters (volume of distribution or renal CL) of vancomycin may change with disease progression due to the changes in organ blood flow or renal insufficiency in critically ill patients. In these situations, AUC estimation after the first dose may not be accurate and may lead to the overestimation of the dose and/or show adverse effects due to drug toxicity. Finally, the virtual PK profiles derived from a pop PK model and Monte Carlo simulation was used to validate the proposed methods. To further reflect its validity and applicability, a clinical study in the patient cohort is ongoing. The data obtained from the clinical study and generated with our proposed methods will be compared using Bayesian approach.

## 5. Conclusions

Vancomycin has a narrow therapeutic window and exhibits high interpatient variability. Therefore, it is essential to establish a reliable method to predict AUC_SS_ of vancomycin with accuracy in individual patient. In this report, we proposed two innovative approaches to guide subsequent dosing of vancomycin using limited sampling strategy following the first dose. With three or four concentration levels obtained after the first dose, the proposed methods can accurately estimate AUC_SS_, thus provide guidance for adjusting the dose from the third dose. The feasibility of the two methods was validated using the virtual PK profiles generated from a pop PK model combined with Monte Carlo simulation. The methods propose newer approaches for the individualized dosing of vancomycin based on two-compartment model. Thus, it could be very beneficial for the precise dose selection for vancomycin.

## Figures and Tables

**Figure 1 pharmaceutics-14-01004-f001:**
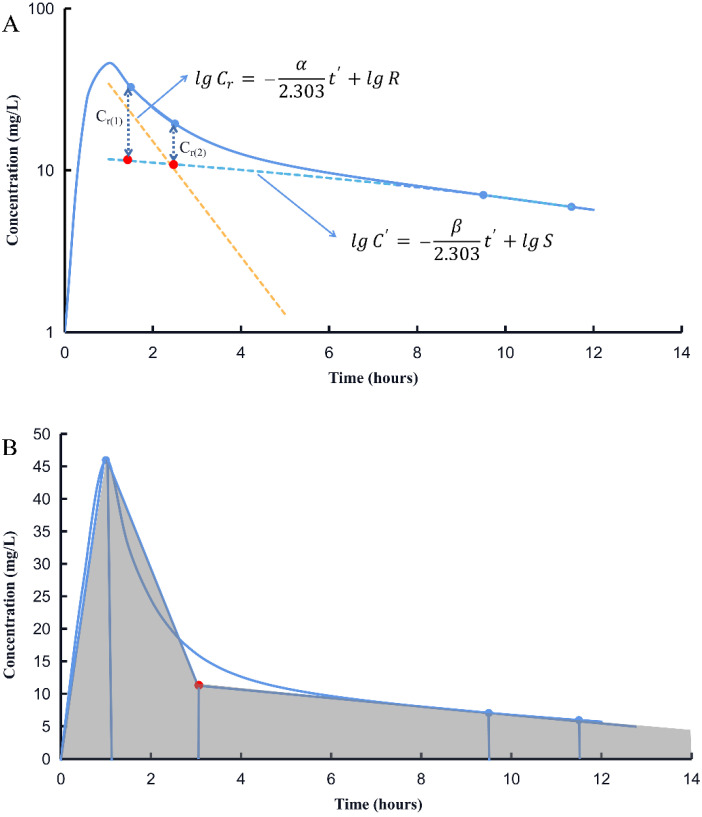
Simulated vancomycin plasma concentration–time profile of a typical patient after 1000 mg intravenous infusion over 60 min fitted to two-compartment PK model. (**A**) Estimation of PK parameters with Method 1 (a mathematical approach using method of residuals). (**B**) Extrapolation of the fourth concentration (red solid circle) using terminal elimination phase and calculation AUC based on linear-trapezoidal rule (sum of shadowed area). The blue points on the curve represent the concentrations at corresponding time.

**Figure 2 pharmaceutics-14-01004-f002:**
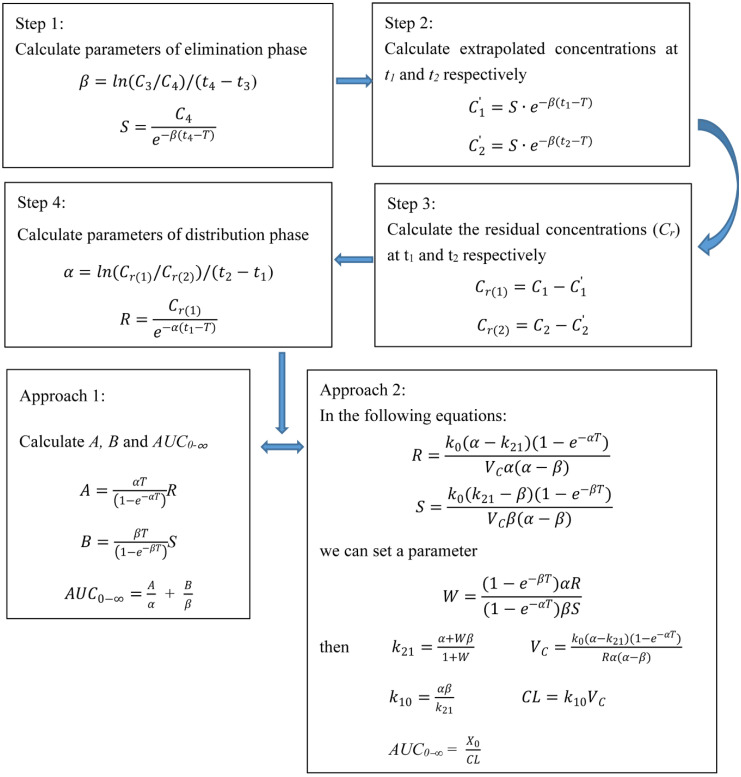
A step-by-step flow-sheet of the equations based on Method 1 to calculate the area under the curve (*AUC*_0–∞_) of vancomycin. α and β are the respective rate constants represent distribution and elimination phases, *Τ* is the infusion time. *C*′ is the extrapolated concentration. *C_r_* is the residual concentration. *V_C_* is the volume of distribution of the central compartment, *k*_21_ is intra-compartmental rate constant, and *A* and *B* are intercepts for distribution and elimination phases. *k*_10_ is the elimination rate constant from the central compartment, *X*_0_ is the dose.

**Figure 3 pharmaceutics-14-01004-f003:**
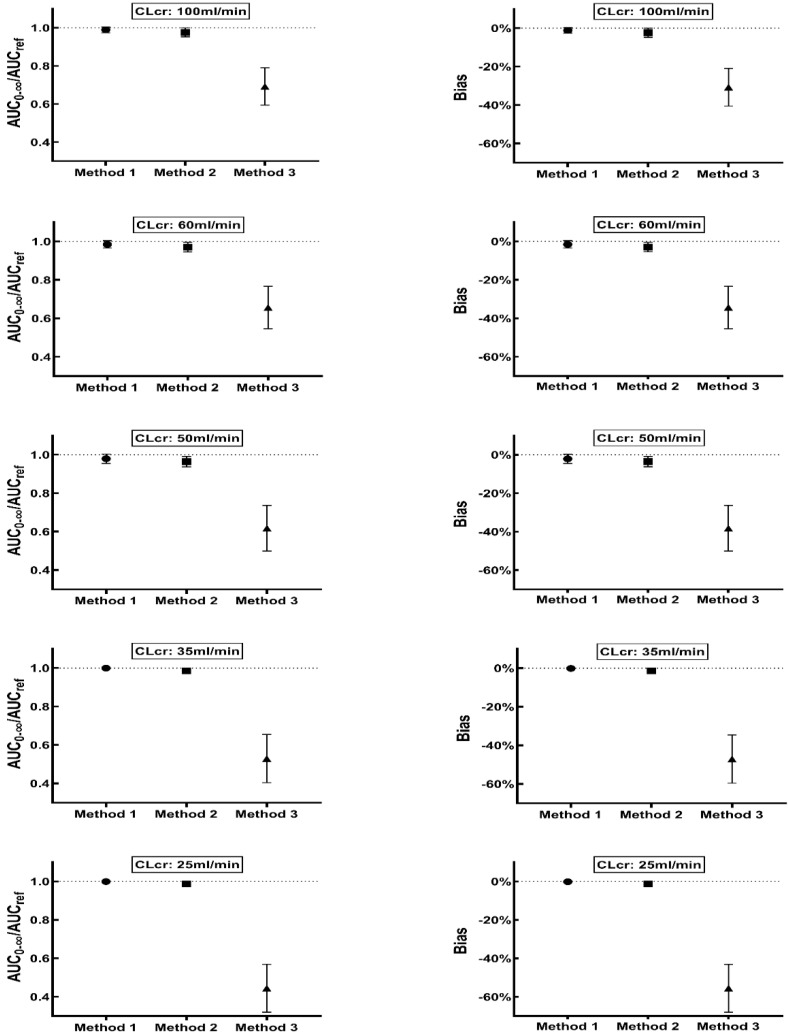
Ratio of *AUC*_0–∞_: *AUC_ref_* (**left panel**) and Bias (**right panel**) calculated using three methods in patients with various degrees of renal function (CLcr value: 100, 60, 50, 35 and 25 mL/min). Data is presented as mean and SD. The solid circle, square, triangle represent the mean value of *AUC*_0–∞_: *AUC_ref_* (**left**) or Bias (**right**) using Method 1, Method 2, and Method 3, respectively.

**Figure 4 pharmaceutics-14-01004-f004:**
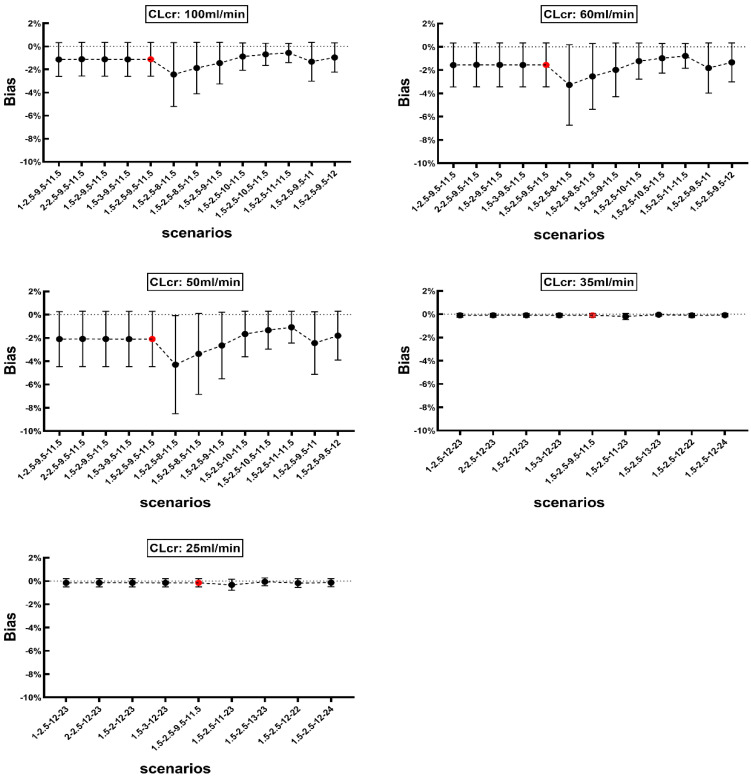
The effect of changing sampling time on bias for *AUC*_0–∞_ calculated using Method 1 in patients with various degrees of renal function (CLcr: 100, 60, 50, 35, 25 mL/min). Red solid circle represents the mean bias when the four sampling time points set per the schedule, namely 1.5, 2.5, 9.5 and 11.5 h post the initiation of intravenous infusion of vancomycin. Black solid circle represents the mean bias with changing one scheduled sampling time while other three remain unchanged. X axis represents respective sampling time schedule for each patient, for example, 1-2.5-9.5-11.5 represents that the four sampling time points set at 1, 2.5, 9.5 and 11.5 h post the initiation of intravenous infusion of vancomycin. Data is presented as mean and SD.

**Figure 5 pharmaceutics-14-01004-f005:**
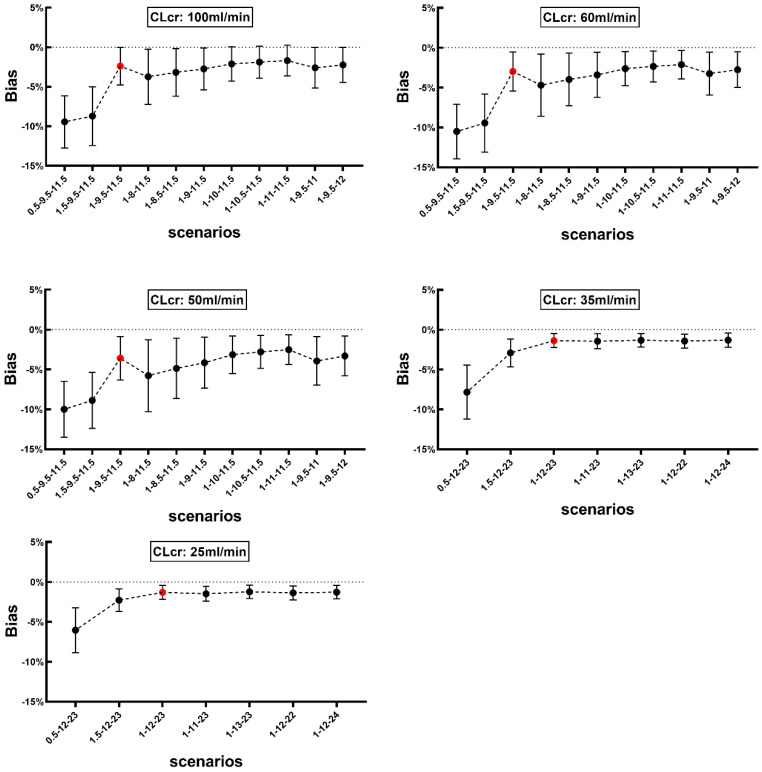
The effect of changing sampling time on bias for *AUC*_0–∞_ calculated using Method 2 in patients with various degrees of renal function (CLcr value: 100, 60, 50, 35, 25 mL/min). Red solid circle represents the mean bias when the three sampling time points set as per schedule, namely 1, 9.5 and 11.5 h after start of vancomycin intravenous infusion. Black solid circle represents the mean bias with changing one scheduled sampling time with the others being unchanged. X axis represents the sampling time schedule. Data is presented as mean and SD.

**Figure 6 pharmaceutics-14-01004-f006:**
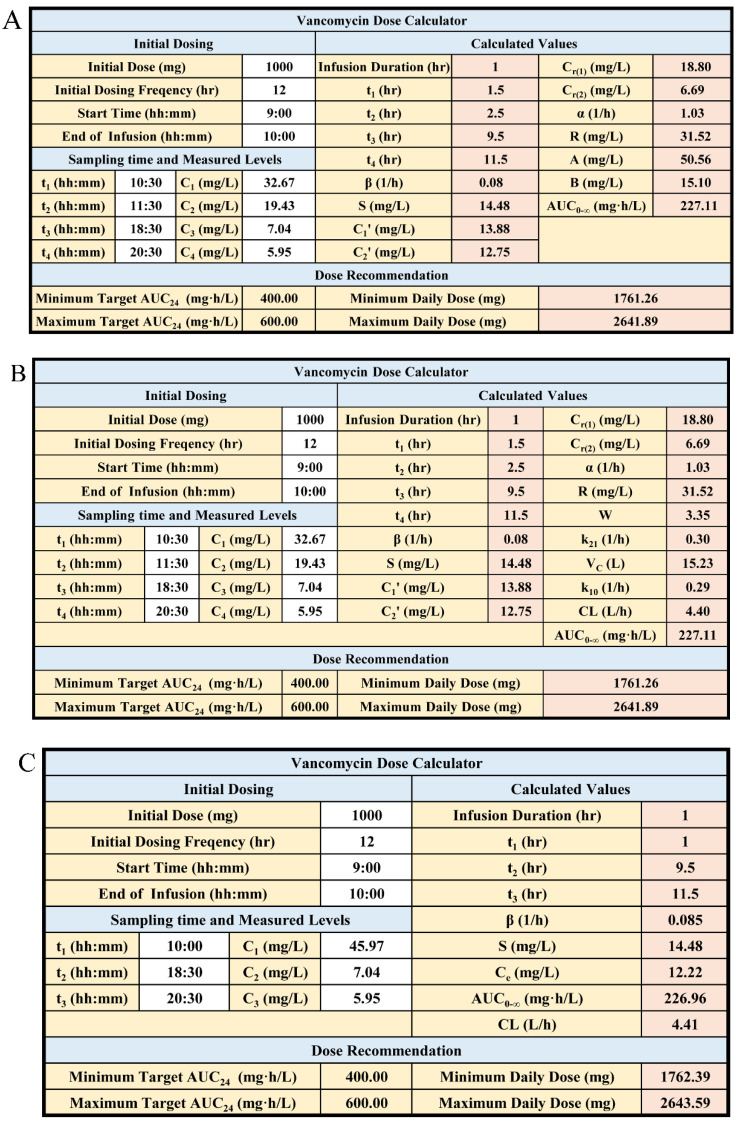
Screenshot of the calculators based on Method 1 and Method 2. (**A**) Method 1, approach 1; (**B**) Method 1, approach 2; (**C**) Method 2. White cells in the upper portion in the table are the patient-specific information. The blocks in pink would be auto populated. After calculating *AUC*_0–∞_, the recommended daily dose to target the minimum and maximum PK/PD index would be displayed in the cell on lower right.

**Table 1 pharmaceutics-14-01004-t001:** Dosing regimen and scheduled sampling time in patients with various degrees of renal function.

CLcr (mL/min)	Time Reaching SS (day)	Dosage (mg)	Dosing Interval (h)	Infusion Rate (mg/h)	Sampling Time (h)	
					Method 1	Method 2
100	15	1000	12	1000	1.5, 2.5, 9.5, 11.5	1, 9.5, 11.5
60	30	750	12	750	1.5, 2.5, 9.5, 11.5	1, 9.5, 11.5
50	30	500	12	500	1.5, 2.5, 9.5, 11.5	1, 9.5, 11.5
35	30	750	24	750	1.5, 2.5, 12, 23	1, 12, 23
25	30	500	24	500	1.5, 2.5, 12, 23	1, 12, 23

SS: steady state; CLcr: creatinine clearance.

## Data Availability

All data included in this study are available from corresponding author upon request.

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
