# Peer review of "Two Innovative Approaches to Optimize Vancomycin Dosing Using Estimated AUC after First Dose: Validation Using Data Generated from Population PK Model Coupled with Monte-Carlo Simulation and Comparison with the First-Order PK Equation Approach"

_pharmaceutics, 2022, doi:10.3390/pharmaceutics14051004_

Round 1

Reviewer 1 Report

This manuscript demonstrated the innovative first-order PK equation approach to optimize vancomycin dosing after first dose. Early steady state AUC estimation was obtained before reaching steady state, and third or fourth dose can be optimized with this approach. Compared with 2-point sampling using 1-compartment model, higher accuracy and lower bias was demonstrated in 3 to 4-point sampling using 2-compartment model.

  1. Title: validation of new first-order PK equation approach should be included in the title. “Two” is not required.
  2. In table 1, authors demonstrated doing regimen according to the renal function. In patients with normal renal function, fixed 1000mg q12h was suggested. However recent guidelines recommended a loading dose (e.g. 1500mg) to increase AUC on day1 or day2.
  3. To increase early clinical response, adequate initial dosing is mandatory before the adjustment of dose by the TDM. The achievement of AUC targets of 400-600 was very low (30-40%) not only in patients with normal renal function but also patients with impaired renal function in this study.
  4. Steady state AUC was evaluated in this study. However, I think AUC on day2 is more important
  5. As you mentioned in the abstract (line 17-19), AUC estimation by first-order PK equation using two samples in one-compartment model might be inaccurate. However, number of  samples required for the estimation is very important in clinical practice. Did you investigated AUC estimation using two samples in two-compartment model? The poor feasibility of measuring three or more-point samples should be considered.
  6. Authors recommended two innovative approach (method 1 and 2) equally. In table 3, there was no significant difference in accuracy and bias between method 1 (4 samples) and method 2 (three samples). Hence authors can recommend method 2 because of less samples.
  7. Is any strength exist in your innovative approach compared with AUC-guided dosing using Bayesian software? Your suggested approach is only for the institutions in which Bayesian software is not available? (as mentioned in line 69-77)
  8. line 449-456: As the evaluation of elimination phase, last 2-point samples (9.5h, 11.5h) and as the evaluation of distribution phase, initial 2-point samples was measured in method 1. Using figure 4, authors exaggerated the importance of exact time sampling at 9.5h and 11.5h. However, only 1 to 2% increase of bias was demonstrated with the shift from 9.5h to 8h in patients with various degree of renal function. Is this a significant difference in clinical setting? Instead, because only one-point sample (1h) was obtained for distribution phase in method 2, time shift from 1h to 0.5h or 1.5h caused a significant bias in patients with q12h administration. A significant bias was also demonstrated in time shift from 1h to 0.5h in patients with 24h administration.
  9. Authors developed the dose calculator tool. How the tool can be accessed? Is it available for users from other countries?

Reviewer 2 Report

Overall Comment:

Liu and colleagues present a study exploring use of explicit pharmacokinetic equations, derived using the method of residuals, to provide AUC-based dosing for vancomycin. This technique benefits from avoiding the “black box” nature of the Bayesian methodologies and can be readily implemented by pharmacists without advanced training in pharmacokinetics. The authors utilized a simulation-based approach to assess the robustness of their method. I believe the author’s work is timely, however I believe several clarifications will help reinforce the limitations of this method to potential readers.

Specific Comments:

  1. One compartment and three compartment models were excluded. As the authors state, selection of the correct model is an underlying challenge with Bayesian dosing strategies. However, assuming a 2CM for vancomycin (though the more common model structure for this drug) could introduce its own bias. In special populations, especially the critically ill, the 2CM assumption may not be valid. The authors should outline clearly in the discussion the limitations of this method.
  2. How were sample times selected? Was an optimal sampling procedure utilized? How sensitive to sample times are these methods and how might the sample times have to be adjusted in the clinical setting?
  3. What is the clinical feasibility of obtaining 3-4 samples within a single dosing interval?
  4. Vancomycin PK is highly dependent on subject body weight in addition to renal function; why were patient weights not included as a covariate on simulations (many models do include weight)
  5. (line 236) The authors should consider including precision evaluation (RMSE) in addition to bias.
  6. (line 229) Why was linear-trapezoidal used to define the reference AUC given that these were simulated from NONMEM and the true clearance values for each simulated subject are known?
  7. (Minor comment, line 155) “adequately large”

Reviewer 3 Report

The manuscript describes two approaches of vancomycin dose optimization based on three or four blood sampling points. The proposed methods, although based on the well-known principles, due to their simplicity and the fact that they do not require any expensive professional software  may be interesting for clinicians. The main drawback of both methods is the necessity to draw several blood samples which is not practical or may be even impossible in many hospitals due to the ethical or logistical issues.

In order to increase the value of the proposed approaches, the presented methods should be compared with the Bayesian method as the most widely used in TDM and the existing software used for dose optimization, e.g. PrecisePK.

The method of AUC calculation in Method 3 should be presented in the Methods section (p. 7).

As vancomycin is often used in critically ill patients, the pharmacokinetic parameters (volume of distribution or renal clearance) of vancomycin  may change with disease progression due to e.g. changes in organ blood flow or renal insufficiency. In this situations, predictions based on the concentrations measured after the first dose may not be correct leading to the overestimation of the dose and drug toxicity.

In order to increase the validity of both methods, they should be verified in real-life conditions.

Reviewer 4 Report

Dear authors,

the article is a novelty in literature, however I recommend to rectify some parts, relying on plagiarism software, because the similarity with other paper is 20% or more. The are some parts in text with double space between words (e.g. line 75, 83, 77, 82, 279..).

Round 2

Reviewer 1 Report

The article was well revised according to the comments.

Reviewer 3 Report

The authors have satisfactorily addressed all my concerns and made the necessary changes to the manuscript.